# The Economic Burden of Localized Prostate Cancer and Insights Derived from Cost-Effectiveness Studies of the Different Treatments

**DOI:** 10.3390/cancers14174088

**Published:** 2022-08-24

**Authors:** David Cantarero-Prieto, Javier Lera, Paloma Lanza-Leon, Marina Barreda-Gutierrez, Vicente Guillem-Porta, Luis Castelo-Branco, Jose M. Martin-Moreno

**Affiliations:** 1Department of Economics, University of Cantabria and Research Group of Health Economics and Health Services Management—Valdecilla Biomedical Research Institute (IDIVAL), 39011 Santander, Spain; 2Fundación Instituto Valenciano de Oncología, 46009 Valencia, Spain; 3NOVA National School of Public Health, NOVA University, 1600-560 Lisbon, Portugal; 4Department of Preventive Medicine and Public Health, Medical School and INCLIVA, Hospital Clínico Universitario de Valencia, University of Valencia, 46010 Valencia, Spain

**Keywords:** economic burden, cost of illness, cost-effectiveness analysis, localized prostate cancer

## Abstract

**Simple Summary:**

Prostate cancer is one of the most frequent and impacting malignant neoplasms for men. In particular, localized prostate cancer has a notably high incidence and prevalence, despite which a solid consensus on treatment and procedure of care has not yet been reached. This article aims to shed light on this challenge by characterizing the economic burden and cost-effectiveness of different treatment strategies for localized prostate cancer after analyzing published comparable data from studies conducted in OECD countries.

**Abstract:**

Prostate cancer has huge health and societal impacts, and there is no clear consensus on the most effective and efficient treatment strategy for this disease, particularly for localized prostate cancer. We have reviewed the scientific literature describing the economic burden and cost-effectiveness of different treatment strategies for localized prostate cancer in OECD countries. We initially identified 315 articles, studying 13 of them in depth (those that met the inclusion criteria), comparing the social perspectives of cost, time period, geographical area, and severity. The economic burden arising from prostate cancer due to losses in productivity and increased caregiver load is noticeable, but clinical decision-making is carried out with more subjective variability than would be advisable. The direct cost of the intervention was the main driver for the treatment of less severe cases of prostate cancer, whereas for more severe cases, the most important determinant was the loss in productivity. Newer, more affordable radiotherapy strategies may play a crucial role in the future treatment of early prostate cancer. The interpretation of our results depends on conducting thorough sensitivity analyses. This approach may help better understand parameter uncertainty and the methodological choices discussed in health economics studies. Future results of ongoing clinical trials that are considering genetic characteristics in assessing treatment response of patients with localized prostate cancer may shed new light on important clinical and pharmacoeconomic decisions.

## 1. Introduction

The current high incidence rate of prostate cancer in the world population is worrisome. Results from different reliable sources such as GLOBOCAN show that around 1.4 million cases of prostate cancer were detected in 2020, being the most common cancer in men. Specifically, Europe is the region of the world with the highest percentage of new cases (37.5%). Moreover, from 1960 to the present day, there has been an upward trend in the rates, which is partly associated with the aging of the population and the increasing use of PSA screening as an early diagnostic test; however, in recent years, this growth has somewhat stalled [1]. In addition, incidence and mortality vary across regions, and this is not only due to biological or environmental exposure factors, but also to accessibility to health services and care options [2].

The main recommendations for the treatment of prostate cancer are found in the clinical practice guidelines published by prestigious and rigorous organizations such as ESMO [3] or the NCCN [4]. A relative lack of consensus on the optimal treatment for localized disease emerges from these guidelines. It is recommended that, given the variety of treatment options and their side effects, men should be offered the opportunity to consult with urologists, radiation oncologists, or medical oncologists, before making their final decisions [3,4]. Several studies, such as Sathianathen et al. [5], identified a favorable evolution of prostate cancer rate trends that were associated with screening practices and the availability of different existing treatments. In the same vein, Sharma [6] showed that these differences were not only found in incidence rates, but also in prostate cancer mortality rates, where more pronounced declines have been generated in high-income countries.

There have been multiple publications analyzing the costs involved in different treatments for prostate cancer. Studies such as the one by Smith-Palmer et al. [7] find that prostate cancer is associated with a significant clinical and economic burden, and that early detection and aggressive treatment leads to improved survival. Moreover, active surveillance remains a safe option for low-risk prostate cancer, even in the long term [8]. On the other hand, Trogdon et al. [9] noted that most of the costs were related to overtreatment and that decreasing the provision of low-value healthcare services for these patients could result in significant healthcare savings. Regarding studies characterizing detailed economic costs, Zaorsky et al. [10] found that the average cost of therapy for a prostate cancer patient is $2800 per month after diagnosis in the United States, with surgery and subsequent office visits accounting for most of the cost. On the other hand, Luengo-Fernandez et al. [11] estimated that the annual costs of prostate cancer are around €8.5 billion for healthcare providers in Europe. Evidently, the cost varies according to the context of care provision, and should be explored further.

Notwithstanding the benefits that different treatments generate for an inclusive society, there is currently no consensus on the cost-effectiveness of prostate cancer treatments. Moreover, such an assessment depends on the specific clinical context, stage of disease, and evidence-based treatment options available. Therefore, research on the objective cost-effectiveness of the different treatments using cost and cost-effectiveness analysis can be useful in reducing healthcare spending and maximizing benefits to the society, patients, and their families. This systematic review aims to uncover the evidence generated by cost-effectiveness studies on the economic burden of localized prostate cancer, a scenario in which we approach the disease with a curative intent, and to compare relevant costs among studies examining major economic drivers. Trying to shed some light on the challenges expressed above, our systematic review focuses on the cost-effectiveness evidence available for therapeutic alternatives that deal with two stages of disease: localized disease and locally advanced disease.

## 2. Materials and Methods

A comprehensive literature search was carried out using the digital platforms of PubMed, Cochrane Library, and Web of Science until May 2022. This was completed to identify the most recent and relevant published evidence regarding cost-effectiveness alternatives for localized prostate cancer. The key words consistently used in all databases were related to “Prostate cancer”, “Cost-effectiveness”, and “Economic evaluation”. The search strategy terms are shown in Table 1. The following inclusion criteria were applied: (1) Articles published in peer-reviewed journals; (2) Papers published within the last 5 years; (3) Studies published in English; (4) Papers examining the cost-effectiveness of therapeutic alternatives for localized and locally advanced prostate cancer. Duplicate articles and those that did not fit the objective of our study were excluded. Search strategies were limited to human studies.

Within this framework, cost estimates were compared among studies with a societal perspective on costs, time-period, and year of price level used. In addition, differences in geographical area and severity group were also considered.

## 3. Results

Our literature search identified 315 publications from the databases considered. Specifically, 256 records were identified in PubMed, 52 in Web of Science, and 7 in Cochrane Library. A total of 96 were duplicates and were therefore eliminated. By carefully and systematically examining the titles according to the eligibility criteria (addressing the cost-effectiveness of different strategies to treat prostate cancer and the economic burden in OECD countries), 50 articles were preliminarily selected. Of these, 37 articles had to be excluded because they did not meet the defined criteria (specifically, we found that 26 did not fit our objective because they only analyzed the diagnosis of the disease or did not address cost-effectiveness analysis; 3 were local studies that self-acknowledged their lack of external validity; 5 were retracted or not directly available; and the remaining 3 were not from peer-reviewed journals). Therefore, a final set of 13 articles was selected for this review.

Figure 1 summarizes the article selection process in diagrammatic form.

Data were extracted from finally selected studies. They covered the three therapeutic approaches for localized prostate cancer: active surveillance, surgery (radical prostatectomy, open, or robotic), or radiotherapy (brachytherapy, external beam RT, or SBRT). Authors manually extracted the main characteristics of the selected articles: author and year, country analyzed, population characteristics, year of costing, currency, therapeutic approach, costs in year of costing, results, perspective, and cost-effectiveness threshold (see Table 2).

Figure 2 includes some data from the included studies: geographical area (Figure 2a) and population included according to risk of prostate cancer typology (Figure 2b). Thirteen studies reporting on the cost-effectiveness of three therapeutic approaches for localized prostate cancer were included in this review. Most of the articles considered here 3analyzed cases from the United States (N = 5), followed by the United Kingdom (N = 3), and the Netherlands (N = 2). The remaining articles correspond to studies conducted in Australia, Canada, and New Zealand. In addition, Figure 2b describes patients according to their level of risk. It should be noted that studies often included patients with different levels of cancer risk. Nine of the articles include information on the cost-effectiveness of treatments for patients with low-risk cancer. On the other hand, five and four articles also include patients with intermediate- and high-risk cancer, respectively. Two of the studies included in the review do not specify the population considered according to disease-associated risk.

## 4. Discussion

This review aimed to summarize the evidence on the economic burden and cost-effectiveness of the main treatments available for localized prostate cancer. Most of the papers included active surveillance, radiotherapy, and radical prostatectomy. In light of the assessments of other treatment alternatives, many papers suggested that there was room for improvement. This would be possible through the adoption of the best clinical practices according to the latest research (including the results of recent clinical trials), which will certainly benefit patients with localized prostate cancer.

For standard economic evaluation, several papers used microsimulation models developed to estimate the cost-effectiveness of different treatment strategies using model inputs based on published literature and economic costs. The variability in these studies explains why there are huge differences in costs depending on the geographical areas analyzed, which may derive from several determinants, including demographic, economic, and social characteristics related to prostate cancer. In addition to these, what seems to condition the results the most are the differences in health service models, which is a determining factor in the heterogeneity of the economic burdens related to localized prostate cancer.

### 4.1. Low-Risk Prostate Cancer

Patel et al. [14] assessed the cost-effectiveness of three active surveillance (AS) strategies for men newly diagnosed with low-risk cancer: (1) AS with transrectal ultrasound-guided biopsy (TRUSGB); (2) AS with multiparametric magnetic resonance imaging (mpMRI) and MRI ultrasound-guided biopsy (MR-TRUSGB); (3) AS with mpMRI without biopsies. They showed that mpMRI with biopsy appeared to be the most cost-effective AS strategy for men with low-risk prostate cancer, including improvements in QoL and cost reduction. It is necessary to mention that MR-TRUSGB would require very specific training. Considering the learning curve and the associated costs of training, this technique may be only cost effective after some time.

On the other hand, Sathianathen et al. [5] evaluated the costs and benefits of different AS follow-up strategies, compared with watchful waiting (WW) or immediate treatment for the same cohort of patients. WW showed an associated cost of US$11,446 and 17,199 per additional quality-adjusted life year (QALY) gained QALY. Immediate treatment costed US$21,819 with 17,382 per QALY. Finally, following up with magnetic resonance imaging (MRI) every 5 years had an associated cost of US$19,850 and 17,572 per QALY. These results show that conservative management of low-risk disease tends to optimize health outcomes and costs, whereas incorporating magnetic resonance imaging (MRI) into surveillance protocols can be cost-effective in some cases, depending on the MRI costs (cost-effective when is done every 5 years with an incremental cost-effectiveness ratio (ICER) of US$92,068 per QALY). Other MRI strategies, although used more frequently, were shown to have ICERs beyond $800,000 per QALY and therefore cannot be advised as cost-effective.

Complementarily, the work reflected in the article by Winn et al. [23] created a simulation model to compare “appropriate imaging” with the status quo. According to the authors, applied imaging, radiation, and surgery had an associated cost of US$409, US$23,145 and US$28,507, respectively. The results indicated that the incremental cost-effectiveness ratio of ideal upfront imaging was less costly (and at the same time slightly more effective) compared with current practice patterns, i.e., simple imaging and interventions according to standard guidelines. Therefore, there are alternatives to improve the efficiency and value of prostate cancer care by reducing costs, and these options can be applied to similar care settings.

Hehakaya et al. [21] examined the necessary relative reduction in complications and the maximum price of 1.5 Tesla MRIs in a simulated cohort of low- and intermediate-localized prostate cancer. Resonance imaging radiotherapy linear accelerator (MR-Linac) was shown to be cost-effective compared with 5, 20, and 39 fractionation schedules of external beam radiotherapy (EBRT) and low-dose-rate (LDR) brachytherapy. According to their results, MR-Linac was cost-effective compared with 20 and 39 fractions EBRT at baseline. However, for MR-Linac to be considered cost-effective compared with 5-fraction EBRT and LDR brachytherapy, substantial reductions in complications must be confirmed or directly offered at a lower cost.

Thus, the above studies suggest a favorable cost-effectiveness ratio of implementing MRI-based imaging or RT treatment for patients with localized prostate cancer, an advantage that becomes more marked if the cost per MRI session is reduced.

Additional studies worth noting included Degeling et al. [20], who estimated the lifetime health and economic outcomes of selecting active surveillance (AS), radical prostatectomy (RP), or radiation therapy (RT) for the initial management of low-risk localized prostate cancer in Australia. The results indicated that AS did not prove to be a cost-effective strategy for lifetime localized low-risk prostate cancer, due to the increased number of patients developing metastatic disease. For them, RT was the dominant strategy that yielded higher QALYs at lower cost. Although the differences compared with RP were small, this small decrease in survival may signify the delay or avoidance of possible complications of treatment. Thus, the patient’s decision emerges once again as fundamental.

Harat et al. [15] compared the cost-effectiveness and QALYs of active monitoring (AM), radical prostatectomy (RP), and external-beam radiotherapy with neoadjuvant hormone therapy (RT). According to their analyses, the costs for these treatment options were $30,378, 18,791, and 15,654 over 10 years for RT, PR, and AM respectively. The incremental cost-effectiveness ratio (ICER) was $6548 for PR over AM and $68,339 for RT over RP, although these incremental costs were below common willingness-to-pay thresholds. More clinical data are needed to better identify patients who can be spared invasive treatments from those who really need it, which could allow for improved cost-effectiveness analyses.

Shanghera et al. [18] examined the lifetime cost-effectiveness of localized prostate cancer treatment according to different subgroups, including low-risk patients. According to the authors, RT was the least costly strategy and generated more QALYs overall, compared with AM (£2455 and 0.08 QALY difference) and RP (£500 and 0.09 QALY difference). Notably, in all analyses in this study (even for patients with low-risk disease), AM was associated with higher costs and a trend toward higher-risk for metastatic (non-curable) disease. They concluded that radiotherapy was the best strategy for low-risk patients.

Finally, Lao et al. [12] compared the cost-effectiveness of active surveillance, watchful waiting, and radical prostatectomy for low risk- prostate cancer patients in New Zealand. WW, AS, and RP had associated costs of NZ$323, 980, and 13,527, respectively. According to the results, WW had lower costs but also lower health outcomes. Moreover, the health outcomes (QALYs) were lower for AS than for RP. On the other hand, authors found differences according to age group. For younger patients (<55), AS was more expensive than RP (and RP more cost-effective than WW). For older patients, RP was more costly than AS, and WW was dominated by the other strategies. From the interpretation of these data, it can be concluded that for men older than 60, AS was cost-effective. The authors concluded that the best strategy in terms of cost-effectiveness depended on patient characteristics and estimated life expectancy.

### 4.2. Intermediate-Risk Prostate Cancer

The aforementioned study by Hehakaya et al. [21] found that MR-Linac, compared with EBRT and LDR brachytherapy could potentially be cost-effective for both low- and intermediate-risk localized prostate cancer.

Dorth et al. [13] carried out a cost-effectiveness analysis, measuring quality-adjusted life expectancy (QALE) and cost between two treatment options for intermediate- to high- risk prostate cancer, in a target population of intermediate-risk patients: (1) Radiation (RT) with androgen deprivation therapy (ADT); (2) Radical prostatectomy (RP) followed by adjuvant RT. They concluded that across all primary and secondary analyses, using a wide-range of assumptions, RT-ADT was the preferred treatment strategy for men with intermediate-risk prostate cancer. It was cost-effective in alternative situations and fell beneath the threshold of $100,000 per QALY.

Moreover, Noble et al. [16] presented an economic evaluation of individual patient data from the ProtecT trial in terms of costs to the UK National Health Service and average QALYs at a 10-year follow-up—a prespecified time point for the primary analysis. Subgroup analyses confirmed that radiotherapy was cost-effective for older men and intermediate-risk disease groups.

Lastly, Shanghera et al. [18] estimated the lifetime cost-effectiveness of managing localized prostate cancer according to different sub-groups, including intermediate-risk patients. According to the authors, RT is the relative least costly strategy and generates more QALYs overall compared with AM (£2455 and 0.08 QALY difference) and RP (£500 and 0.09 QALY difference). When analyzed according to age groups, prostatectomy had the greatest net benefit for men younger than 65 years and radical radiotherapy for those older than 65 years, but sensitivity analysis showed considerable uncertainty in both results.

### 4.3. High-Risk Prostate Cancer

Although Noble et al. [16], concluded that radiotherapy was cost-effective for older men and high-risk disease groups in their analysis of the previously mentioned ProtecT trial, Shanghera et al. [18] reported that radical prostatectomy appeared to be the most cost-effective for high-risk prostate cancer patients.

Subsequently, Dorth et al. [13] conducted a cost-effectiveness analysis, quality-adjusted life expectancy (QALE), and a comparison of the cost between two treatment options for intermediate- to high-risk prostate cancer. On the one hand, radiation (RT) with androgen deprivation therapy (ADT) has always been one of the mainstays of treatment for high-risk prostate cancer, its benefit deriving from both improved local control and inhibition of micrometastatic disease. On the other hand, radical prostatectomy (RP) followed by adjuvant RT has been shown to be well-tolerated by patients and prolongs biochemical recurrence-free survival compared with radical prostatectomy alone in patients with positive margins or extracapsular extension. From the integration of the above, the authors concluded that across all primary and secondary analyses, and using a wide-range of assumptions, RT-ADT was the preferred treatment strategy for men with intermediate- to high-risk prostate cancer.

Finally, Winn et al. [23] created a state-transition microsimulation model to understand changing population-level patterns of imaging among men with incidental prostate cancer. The results indicated that when only high-risk men were prioritized for imaging tests compared with the status quo (real-world practice from the SEER-Medicare database), both the population rate of imaging tests and average per-person expenditure on imaging tests declined. The incremental cost-effectiveness ratio of ideal upfront imaging was less costly and slightly more effective compared with current practice patterns, that is, guideline-concordant imaging. This again provides knowledge that can be applied to comparable care settings, which could improve the efficiency and value of care provided to prostate cancer patients.

### 4.4. Unspecified Risk

We retrieved two articles that did not specify the associated risk of disease, but provided information of interest. Parackal et al. [17] determined the cost-effectiveness of robot-assisted radical prostatectomy (RARP) from a Canadian public payer’s perspective. At a willingness-to-pay threshold of $50,000 and 100,000 per QALY gained, the authors suggested that RARP is a cost-effective treatment option compared with ORP. It is worth noting that robot surgery is a controversial subject due to the high costs associated with the acquisition of the robot itself, plus the necessary training, application, and maintenance. The costly nature of these robotic surgeries tends to force the careful evaluation of clinical histories, particularly to specific comorbidities on a patient-by-patient basis [24,25]. In addition, some authors argue that a minimum number of surgeries must be performed for the introduction of this option to be worthwhile [26]. This conclusion has been made more drastic in the case of the Health Quality Ontario study [27], which suggested that there is no clear evidence on the additional health benefits of using a robot compared with traditional approaches.

Meanwhile, Schumacher et al. [19] assessed the required toxicity reduction to justify the added costs of MRI-guided radiotherapy (MR-IGRT) over CT-based image guided radiotherapy (CT-IGRT) in localized prostate cancer treatment. They showed that slight toxicity reductions (7–14%) are required for stereotactic body radiotherapy (SBRT) to be cost-effective whereas conventional radiotherapy requires toxicity reductions of 50 and 94% to be cost-effective.

### 4.5. Overall Appraisal of the Existing Evidence and Limitations of the Available Data and of Current Review

All in all, the summarized results indicate that the greater the severity of prostate cancer, the higher the associated cost, mainly due to the need for more intensive treatments. Moreover, the total cost of prostate cancer and the economic burden involved appear to be higher in the United States than in European countries. This can be explained by the fact that direct health care costs (and direct non-medical ones) are larger in the United States, even after accounting for greater indirect costs in Europe due to subsidized absenteeism. These factors affect the cost-effectiveness analyses and the comparability of results.

The time horizon of these models is usually until the death of all patients, so that the effect of follow-up time and additional simulated events on model results can be assessed. An advantage of microsimulation models over a state-transition Markov model is the ability to model and follow individual events.

Researchers should make new efforts to estimate the economic burden of prostate cancer and its cost-effectiveness, even during this era of COVID-19. This will hopefully provide new insights that will enable better decision-making on cost-effective treatment options and public health priorities for prostate cancer. Moreover, the treatment intention for localized prostate cancer is curative and a great majority of patients are fortunately long-term survivors. Thus, a longer follow-up and broader societal health economic perspective (e.g., work absence, impact on caregivers and relatives, long-term follow-ups and toxicity, etc.) should always be considered in future studies. Furthermore, several treatments and outcomes for patients with localized prostate cancer harboring different genetic features (e.g., BRCA mutations, MSI-H, etc.) are being assessed in various clinical trials, and may influence clinical and pharmacoeconomic decisions in the future [28].

This literature review has systematically followed verifiable steps, providing up-to-date empirical evidence on the cost-effectiveness of different prostate cancer treatment strategies in OECD countries. Although we believe in the relevance of the main results of this research, we must make explicit the potential limitations of this review. Among them, one limitation is that we have only included OECD countries. This was decided to ensure relative compatibility between countries, in accordance with the usual practice of comparative studies. To mitigate this limitation, and for better stratification and comparability, we presented our analysis by clinically meaningful tumor staging. Secondly, perfect comparisons cannot be made because of the differences between countries and their health systems, or the costs included in their cost-effectiveness analyses. Moreover, some studies point to the importance of reducing the toxicity of radiotherapy, which would increase its cost-effectiveness. From another perspective, angiogenesis appears to play a crucial role in the evolvement of prostate cancer; although antiangiogenic therapies have not demonstrated significant clinical benefit for cancer patients so far, they may be a potential resource that merit further investigation [29,30]. Taking all these factors into account, both patients and caregivers will need to stay updated with increasing pace, especially as genetic markers become available to stratify populations and allow an approach that is more in line with the concept of precision medicine. For now, we must insist on studies that better characterize the prediction of treatment impact on the evolution of prostate cancer or perform a cost-effectiveness analysis of treatments that are currently being introduced, such as the use of protons, classic photon therapy, rectal spacer placement, or the application of gold anchor markers, as there is still no clear consensus on their cost-effectiveness.

## 5. Conclusions

This study provides updated evidence related to the cost-effectiveness of the different clinical strategies for the treatment of localized prostate cancer at different stages of severity.

Various scientific reports have shown that MRI-based imaging or MRI-guided RT, when compared with other imaging modalities, have a favorable cost-effectiveness ratio for patients with localized prostate cancer. Moreover, this option could be further consolidated over time as we gain greater clinical experience and the costs of technology are reduced.

For the population with low-risk localized prostate cancer, based on results from different studies, SA was not shown to be cost-effective overall, although there is some evidence (such as the study by Lao et al. [12]) showing that SA may be cost-effective in men older than 60 years. In any case, the best strategy in terms of cost-effectiveness depends on patient characteristics and more clinical-based data is needed to better identify the patients that can be spared from invasive treatments.

For patients with intermediate- and high-risk prostate cancer, both RT (+ADT) and PR were cost-effective. Importantly, two studies (Noble et al. [16] and Shanghera et al. [18]) showed that the benefit with RT was more marked in older patients.

Regarding surgical strategies, at the moment, there is no clear evidence on the additional health and cost-effectiveness benefits of using a robot compared with traditional approaches.

The conclusions of this analysis are intended to be a basis for future studies on the economic burden or cost-effectiveness of prostate cancer, to better inform clinical practice. However, an increase in cost is, not surprisingly, intimately related to an increase in the severity of prostate cancer. Moreover, the dimension provided by the social perspective is currently more important in Europe than in the United States, but we believe that it should constitute a basic pillar in future assessments for all populations.

Finally, the landscape of diagnosis and treatment options for patients with prostate cancer is evolving very rapidly. For instance, PET-PSMA imaging is becoming more commonly used for diagnosis and treatment guidance, novel biomarker-driven systemic treatments (e.g., PARP-inhibitors or immune checkpoint inhibitors) have shown promising results for a subset of patients with prostate cancer, and new radiotherapy modalities (such as photon therapy) are being tested. In addition, results of ongoing clinical trials that are considering genetic characteristics when assessing treatment response for patients with localized prostate cancer may become key ingredients to shed new light on clinical and pharmacoeconomic decisions. The work ahead is daunting, but the opportunity to provide better care for prostate cancer patients is well worth the research effort.

## Figures and Tables

**Figure 1 cancers-14-04088-f001:**
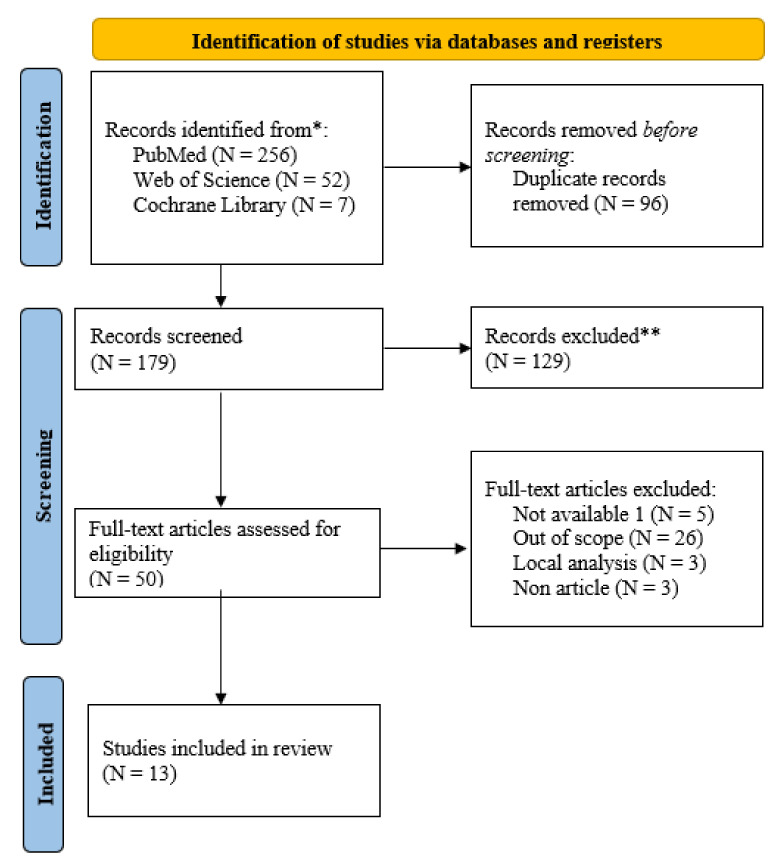
Flow diagram of the paper selection process. * publications from the databases considered. ** after considering the eligibility criteria described in the text.

**Figure 2 cancers-14-04088-f002:**
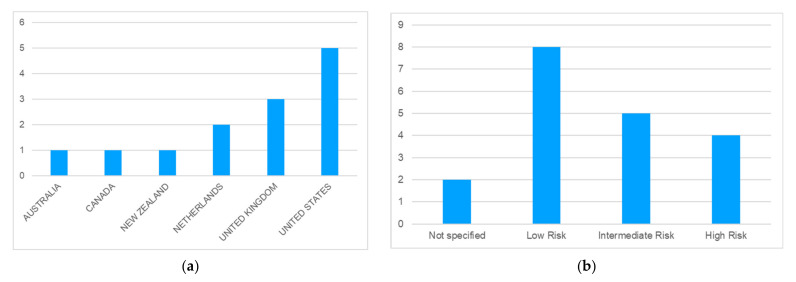
(**a**) Distribution of the articles according to the country analyzed (N = 13). (**b**) Distribution of the articles according to disease-associated risk.

**Table 1 cancers-14-04088-t001:** Search strategy for study selection from PubMed, Cochrane Library, and Web of Science.

#	Search Term
PubMed	
#1.	Prostate cancer [Title]
#2.	Cost-effectiveness [Title/Abstract]
#3.	Economic evaluation [Title/Abstract]
#4.	Limit to: journal article; year of publication: last 5 years; English; Humans subjects, free-full text.
Cochrane Library	
#1.	Prostate cancer [Title]
#2.	Cost-effectiveness [Title/Abstract/Keyword]
#3.	Economic evaluation [Title/Abstract/Keyword]
#4.	Limit to: year of publication: last 5 years.
Web of Science	
#1.	Prostate cancer [Title]
#2.	Cost-effectiveness [Title]
#3.	Economic evaluation [Title]
#4.	Limit to: journal article; year of publication: last 5 years; English; Health Care Sciences Services & Economics; Web of Science Core Collection.

Source: Author’s own elaboration.

**Table 2 cancers-14-04088-t002:** Mapping of studies included in the analysis (N = 13).

Article	Country	Year of Costing-Currency	Population Characteristics	Treatment Approach *	Costs	Perspective	QALY	Cost-Effectiveness Threshold
Lao et al. (2017) [12]	New Zealand	2012/2013 NZ Dollars	Low-risk cancer	AS, WW, RP	RT (13.527); AS (980); WW (323).	Ministry of Health	-	-
Dorth et al. (2021) [13]	USA	2013 US Dollars	Intermediate- and high-risk cancer	RT, RP	RT: Intermediate-risk ranges 26,900/27,500–31,300/33,100. High-risk: 65,300/75,600 RP: Intermediate-risk ranges 20,400/21,300–22,800/24,000. High-risk: 28,500/31,400	Payer	RT: Intermediate-risk ranges 9,78/12,07–9,31/11,45. High-risk: 9,05/11,15 RP: Intermediate-risk ranges 8,89/10,92–8,78/10,82. High-risk: 7,91/9,66	US$100,000/QALY
Patel et al. (2018) [14]	The Netherlands	2016 Euros	Men with low-risk prostate cancer	AS	Unit costs: AS (€100 per year); TRUSGB (€481); mpMRI (€317); MR-TRUSGB (€481); RP (€12,800); RT (€4035); Palliative care (€13,780). Mean costs per man screened: AS TRUSGB (€5150); mpMRI without biopsy (€5994); AS mpMRI with biopsy (€4848).	Healthcare	QALYs were higher for AS mpMRI with biopsy compared with AS TRUSGB (18.67 vs. 18.66) and lower for AS mpMRI without biopsy compared with TRUSGB (18.27 vs. 18.66).	$50,000/QALY
Sathianathen et al. (2019) [5]	USA	2017 US Dollars	Men with low-risk prostate cancer	(1) WW; (2) RP; (3) AS.	Strategy cost: WW (11,446); MRI (20,812). Intermediate treatment (21,819)	Health sector	Intermediate treatment is dominated by WW. MRI every 5 years has an ICER of 92,068. More frequent: not cost-effective.	ICER less than $100,000
Harat et al. (2020) [15]	USA	2008 US Dollars	Low-risk	AS, RP, RT	The mean cost for AM, PR, and RT were $15,654, $18,791, and $30,378	US healthcare payer	The mean QALYs for AM, PR, and RT were 6.96, 7.44, and 7.9 years, respectively.	$50,000 per QALY
Noble et al. (2020) [16]	UK	2015 UK Pounds	Low-, intermediate-, and high-risk	AS, RP, RT	Active monitoring had lower adjusted mean costs (£5913) than radiotherapy (£7361) or surgery (£7519).		Adjusted mean QALYs were similar between groups: 6.89 (active monitoring), 7.09 (radiotherapy), and 6.91 (surgery). Active monitoring had lower adjusted mean costs (£5913) than radiotherapy (£7361) and surgery (£7519).	£20,000 per QALY
Parackal et al. (2020) [17]	Canada	2019 Ca Dollars	Stage I and II	Robotic RP	Total cost of RARP and ORP were $47,033 and $45,332, respectively	Public payer	Total estimated QALYs were 7.2047 and 7.1385 for RARP and ORP, respectively. The estimated incremental cost-utility ratio (ICUR) was $25,704.	CA$50,000 and CA$100,000/QALY
Sanghera et al. (2020) [18]	UK	2015 UK Pounds	Low-, intermediate-, and high-risk	RT, RP	-	NHS	RT generated the greatest net monetary benefit (£293,446 [95% CI £282,811 to 299,451] by D’Amico and £292,736 [95% CI £284,074 to 297,719] by Grade group 1).	£27,000 per QALY
Schumacher et al. (2020) [19]	USA	2019 US Dollars	-	RT	Cost per patient. Conventional radiotherapy (39 fractions): CT-IGRT ($8707); MR-IGRT ($18,836). SBRT (5 fractions): CT-IGRT ($5357); MR-IGRT ($6816).	Healthcare	-	$50,000/QALY and $100,000/QALY
Degeling et al. (2021) [20]	Australia	2020 A Dollars	Low-risk	AS, RP, RT	A$17,912 for AS, 15,609 for RP, and 15,118 for RT	Public Payer	QALYs were 10.88 for AS, 11.10 for RP, and 11.13 for RT. RT had a 61.4% chance of being cost-effective compared with 38.5% for RP and 0.1% for A	A$20,000/QALY
Hehakaya et al. (2021) [21]	The Netherlands	2019 Euros	Simulated 1000 men with low- and intermediate-risk localized prostate cancer	RT	Total cost per patient: EBRT 5 fractions (1635); EBRT 20 fractions (6530); EBRT 39 fractions 12,740); LDR brachytherapy (4585); MR-Linac (6460)	Dutch healthcare	Incremental QALYs: EBRT 5 fractions (+0.06); EBRT 20 fractions (+0.23); EBRT 39 fractions (+0.11); LDR brachytherapy (+0.03)	€80,000/QALY
Labban et al. (2022) [22]	UK	2020/2021 UK Pounds and US Dollars	Lower risk of biochemical recurrence (BCR)	Robotic RP	The total direct 10-year costs of RARP were estimated at £13 247 (US $17,443); those of LRP, at £15,032 (US $19,794); and those of ORP, at £12,721 (US $16,751). Robotic-assisted radical prostatectomy had the highest surgical equipment cost at £2775 (US $3654), followed by LRP at £1360 (US $1791), and ORP at £638 (US $840)	NHS	Compared with LRP, RARP cost £1785 (US $2350) less and had 0.24 more QALYs gained; thus, RARP was a dominant option compared with LRP. Compared with ORP, RARP had 0.12 more QALYs gained but cost £526 (US $693) more during the 10-year time frame, resulting in an ICER of £4293 (US $5653)/QALY	£30,000 [US $39,503]/QALY)
Winn et al. (2022) [23]	USA	2017 US Dollars	Simulated prostate cancer cases in men aged 65 and older.	IM	Treatment specific costs: Imaging (409); Radiation (23,145); Surgery (28,507); Systemic therapy (77,035); Annual costs (2769)	-	QALYs: Status quo imaging (11,075); Appropriate imaging (11,075).	-

* WW: Watchful Waiting; RP: radical Prostatectomy; RT: Radiotherapy; AS: Active Surveillance, IM: Imaging. QALY: cost per additional quality-adjusted life year gained.

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
