# Peer review of "The Economic Burden of Localized Prostate Cancer and Insights Derived from Cost-Effectiveness Studies of the Different Treatments"

_cancers, 2022, doi:10.3390/cancers14174088_

Round 1

Reviewer 1 Report

In their review article, authors aimed to investigate the economic burden of localized prostate cancer and give insights derived from cost-effectiveness studies of the alternative treatments. They performed a meta-analysis and systematic review of 13 studies. The number of included studies is adequate, and I have read the review with great interest. Overall, this is a thorough review. Introduction is reasonable, stats and concomitant results are sound. The discussion is coherent and interprets the findings with current literature. However, some minor details need to be addressed to strengthen the discussion and the overall manuscript. 1) In the introduction you give overtreatment of early stage PC as a main reason for the high cost of prostate cancer (l.60). In this regard, favorable outcome of patients with incidental PC upon TURP was recently shown (PMID: 35053530), advocating that active surveillance is a safe option for low-risk PC patients. 2) As this is a review, please state the exclusion criteria more clearly. If I am not mistaken, you reviewed 50 studies, and only included 13. How were 26 “out of scope”? Please elaborate.

Author Response

Dear Reviewer,

We are very grateful for your kind comments and valuable suggestions. We have taken them all into account, taking this opportunity to inform you of all the changes made in the revised version of the manuscript (at your suggestion, and taking the opportunity to introduce other minor corrections), according to the attached systematized report to your attention. Moreover, we are also sending through the Cancers-mdpi platform two copies of the manuscript (one is the revised manuscript as a clean copy; and the other file related to the revised manuscript highlights the main changes introduced [track changes]).

We hope that we have adequately addressed your comments and improved the manuscript with respect to the initial version, and we look forward to your review, for which we thank you in advance.

With kind regards

Reviewer 2 Report

The authors provide a review regarding interesting topics for clinicians dealing with PC.

Points to be considered:

1) The rationale of why the authors came up with this review.

2) What is the information that is not exactly available that motivated the authors to come up with this information. What are the current caveats and how do the authors highlight the current research in answering them? If not they need to address in future directions.

3) The authors could provide a little more consideration of genomic directed stratifications in clinical trial design and enrollments.

4) The underlying message here is that more precision and individualized approaches need to be tested in well designed clinical trials – a challenge, but I would be interested in their perspective of how this might be done.

5) The authors also deal with metastatic diseases. Bone is one of the preferred site of secondary localization in PC. As is now well known, tumors grow and evolve through a constant crosstalk with the surrounding milieu, and emerging evidence indicates that several gatekeepers and immune-tolerogenic environments frequently occur simultaneously in response to this crosstalk. Accordingly, strategies combining anti-tumor therapy and immunotherapy seem to hold the promise to tip the balance of the cancer niche and improve patient outcomes (Please refer to PMID: 32064051 or similar manuscript and expand the introduction/discussion sections)

6) The authors need to highlight what new information the review is providing to enhance the research in progress.

Author Response

(The authors gave the same response as above.)

Reviewer 3 Report

Generally, this review is somewhat unsatisfactory because "hard" quantitative measures are difficult to assess across multiple countries - the authors tried the impossible anyway, to compare issues that are notoriously difficult to compare, especially quantitatively. This is of course an inherent problem of this kind of studies - that they are difficult to compare across different countries and areas and patient groups, etc etc. I don't really see a solution to this issue; maybe the problem cannot even be resolved in a satisfactory fashion, as the final conclusions (chapter 5 "conclusions") remain rather vague and dont add much to earlier studies that have come to similarly limited conclusions. 

 One may even argue: that maybe its not productive to even attempt such cross-national comparisons? Or should you rather stay in one system and try to get really strong numbers and statistics? Maybe thats then easier to compare to other countries; and one conclusion might be that "country X has solved the problem more effectively than country Y and Z".

Especially since the standards of treatment are also different between these countries (despite them all being "Western" countries that are part of the OECD). Maybe sticking to one system is more conclusive and helpful than cross-system/country comparisons? 

So some readers may be left unsatisfied by the article, and maybe thats part of the message - if this is the case, it should be discussed in this fashion, too. 

One element I see missing is the precise overlap of the question(s) addressed in this review, with the question of quality of life (QoL) in prostate cancer patients. And, more specifically, how different strategies such as"watchful waiting" or active surveillance versus more aggressive therapies may impact the QoL of these patients: how would this be measured, what are the main limiting factors; how does it relate to the QALY measures used here; and arent these also very different from country to country?  There are many articles that target this topic, and obviously, this cannot be covered in all detail here. But it is potentially an important component of the most relevant considerations related to treatment, economic burden, and effectiveness. Here, however, QoL issues are barely mentioned (e.g. on page 8, line 165), apart from being covered in reference 14. Of course, QoL is also assessed as part of the QALY statistics, or "quality-adjusted life year", but how exactly is this defined? Are these measures and standards also different from country to country? So the entire issue is just not clearly defined and while I don't have a recommendation on how this could be improved, it still feels like this isn't covered enough. 

smaller things: 

The title is somewhat misleading. Especially, the term "alternative treatments" made me first believe that this concerns treatments with non-approved forms of therapy, or "alternative therapies" in a classical sense, in contrast to the standard-of-care clinical therapies. 

the frequencies of PrCa cited via Globocan in the introduction are a bit off; first of all it appears that lung cancer is about as frequent as PrCa, or even more common;  and the numbers for the Americas are missing in the listings. 

Author Response

Dear Reviewer,

We are grateful for your comments and valuable suggestions. We have taken them all into account, taking this opportunity to inform you of all the changes made in the revised version of the manuscript (at your suggestion, and taking the opportunity to introduce other minor corrections), according to the attached systematized report to your attention. Moreover, we are also sending through the Cancers-mdpi platform two copies of the manuscript (one is the revised manuscript as a clean copy; and the other file related to the revised manuscript highlights the main changes introduced [track changes]).

We hope that we have adequately addressed your comments and improved the manuscript with respect to the initial version, and we look forward to your review, for which we thank you in advance.

With kind regards

Round 2

Reviewer 2 Report

This reviewer still has the feeling that the introduction discussion sections might benefit from a boosting of discussion about the fact that prostate 
Tumor angiogenesis is sustained by multiple growth factors, including vascular endothelial growth factor (VEGF), placenta growth factor (PlGF), stromal-derived factor-1 (SDF-1), platelet-derived growth factor (PDGF), and fibroblast growth factor (FGF). Accumulating evidence indicates that angiogenic growth factors (AGFs) are involved in tumor growth and progression via multiple mechanisms that go beyond their angiogenic role. Indeed, within the tumor microenvironment (TME), AGFs produced by tumor and inflammatory cells create a tolerogenic milieu that allows the escape of cancer cells from immune recognition and elimination. Therapeutic strategies targeting AGFs to restore vessel normalization have been found to improve antitumor immunity and, in turn, immune-mediated mechanisms can regulate the response to antiangiogenic therapy. Still, immunotherapy is effective only in a fraction of cancer patients, and combined antiangiogenic and immunotherapy approaches have not always yielded the expected clinical results. An in-depth characterization of the complex interplay between AGFs, cancer cells and immune cells in the TME is urgently needed to identify new targets that could promote or hinder the antitumor immune response. Nonetheless, the reviewing process is also subjective. The last decision is for the Editor.

Author Response

Dear Reviewer,

Thanks for your comments and suggestions.

The importance of angiogenesis in the development of prostate cancer is irrefutable, as in other tumors, but the purpose of our review was to examine the cost-effectiveness of the disease and its treatments, and, at present, antiangiogenesis has not shown any practical role in localized prostate cancer, we wanted to incorporate your perspective into account. The changes made in the revised version of the manuscript are described in detail in the attached report to your attention. Moreover, we are also sending through the Cancers-mdpi platform two copies of the manuscript (one is the revised manuscript as a clean copy; and the other file related to the revised manuscript highlights the main changes introduced [track changes]).

We hope that we have adequately addressed your comments and improved the manuscript with respect to the initial version, and we look forward to your positive review, for which we thank you in advance

With kind regards,
